# Preparation and Characterization of Octenyl Succinate β-Cyclodextrin and Vitamin E Inclusion Complex and Its Application in Emulsion

**DOI:** 10.3390/molecules25030654

**Published:** 2020-02-04

**Authors:** Dongmei Ke, Wenxue Chen, Weijun Chen, Yong-Huan Yun, Qiuping Zhong, Xiaotang Su, Haiming Chen

**Affiliations:** 1College of Food Sciences & Engineering, Hainan University, 58 People Road, Haikou 570228, China; 2Chunguang Agro-Product Processing Institute, Wenchang 571333, China; 3Guangdong Association of Circular Economy and Resources Comprehensive Utilization, Guangzhou 510095, China

**Keywords:** octenyl succinic β-cyclodextrin, vitamin E, inclusion complex, emulsifying property, oxidation stability

## Abstract

Vitamin E (VE) and β-cyclodextrin (β-CD) can form an inclusion complex; however, the inclusion rate is low because of the weak interaction between VE and β-CD. The results of a molecular docking study showed that the oxygen atom in the five-membered ring of octenyl succinic anhydride (OSA) formed a strong hydrogen bond interaction (1.89 Å) with the hydrogen atom in the hydroxyl group of C-6. Therefore, β-CD was modified using OSA to produce octenyl succinic-β-cyclodextrin (OCD). The inclusion complexes were then prepared using OCD with VE. The properties of the inclusion complex were investigated by Fourier-transform infrared spectroscopy (FT-IR), ^13^C CP/MAS NMR, scanning electron microscopy (SEM), and atomic force microscopy (AFM). The results demonstrated that VE had been embedded into the cavity of OCD. Furthermore, the emulsifying properties (particle size distribution, ζ-potential, and creaming index) of the OCD/VE inclusion-complex-stabilized emulsion were compared with that stabilized by β-CD, OCD, and an OCD/VE physical mixture. The results showed that the introduction of the OS group and VE could improve the physical stability of the emulsion. In addition, the OCD/VE inclusion complex showed the strongest ability to protect the oil in the emulsion from oxidation. OCD/VE inclusion complex was able to improve the physical and oxidative stability of the emulsion, which is of great significance to the food industry.

## 1. Introduction

Lipid oxidation is one of the major deteriorative chemical changes that can decrease the quality and safety of products such as milk, cream, coffee whiteners, cream liqueurs, mayonnaise, salad dressings, cheese, spreads, yoghurts, and infant formula, especially in their emulsified forms [1,2,3]. Because of the larger contact surface area between oil droplets and oxidants (oxygen, free radicals, and chelating metals), the lipid in the emulsion is much more susceptible to oxidative deterioration than bulk oil [4]. In order to solve this problem, antioxidants (α-tocopherol, ascorbyl palmitate, carnosol, Trolox, and rosmarinic acid) are directly added to the decentralized food system [3]. On the basis of the polarity, the antioxidants are dispersed evenly in the oil droplets or the continuous phase; however, the very starting point of the oxidation process is the oil–water interface [5]. In other words, a sufficient amount of the antioxidant is needed at the oil–water interface to prevent oxidative species (oxygen, free radicals, transition metals, and lipolytic enzymes) contacting the oil.

An emulsion is a thermodynamically unstable system because of the positive free energy needed to increase the surface area between the oil and water phases. To form a kinetically stable emulsion, emulsifiers must be added before homogenization. Surface-active emulsifiers can adsorb to the oil–water interface, forming a protective membrane which, although its thickness is only a few nanometers, plays important roles in enhancing the physical stability of emulsion droplets and delaying lipid oxidation processes by acting as a barrier to the penetration and diffusion of oxidizing agents [6]. Therefore, lipid oxidation may be controlled by adjusting the properties of the membranes by altering their surface activity, charge, and strength. In addition, increasing the antioxidant capacity of the emulsifier by chemical modification or physical adsorption can also improve the oxidative stability of emulsions. Many emulsifier molecules (e.g., Tweens, pectin, and metalloproteins) contain sugar or amino acid moieties that may also act as free-radical scavengers. In our previous study, hydrophobic ferulic acid was introduced to the pectin molecular via an enzymic catalytic reaction, and the surface activity and the antioxidant ability of the reaction product were enhanced [3]. In addition, the surface charge of emulsion droplets plays an important role in their oxidative stability. Transition metals, especially iron, are basic components in food systems, and they can catalyze the oxidation of dispersed lipids at the oil–water interface [7,8].

Octenyl succinic anhydride (OSA) is permitted for food applications as an additive to the maximum addition level of 3%. Esterification of starch with OSA provides hydrophobic domains (-CH_2_-CH_2_-) and charges (COO^−^), enhancing the emulsifying ability of starch [9]. OSA has also been introduced to pectin [10], hyaluronic acid [11], gum arabic [12], and konjac glucomannan [13]; the surface activity of the OSA-modified polysaccharides is improved. In addition, OSA-β-cyclodextrin was prepared in our previous study and the emulsion stabilized by OSA-β-cyclodextrin was more stable than that stabilized by native β-cyclodextrin [14]. Interestingly, the substitution of OSA influenced the loading capacity and release rate of active small molecule substances such as β-carotene [15]. VE is usually used in oil as a natural antioxidant and diet supplement for protecting against aging [16], coronary disease [17], and cancer [18]. According to the molecular docking results, the interactions (e.g., hydrogen bonds and hydrophobic interactions) between VE and β-cyclodextrin were very weak. VE has been successfully loaded into nanoparticles of modified β-cyclodextrin [19].

Therefore, in the present study, an OSA-modified β-cyclodextrin/VE (OCD/VE) inclusion complex was prepared and characterized by Fourier-transform infrared spectroscopy (FT-IR), ^13^C CP/MAS NMR, scanning electron microscopy (SEM), X-ray diffraction (XRD) and atomic force microscopy (AFM). In addition, the physical and oxidative stability of the emulsions were evaluated via zeta potential, particle size distribution, and 2-thiobarbituric acid reactive substance (TBARS). This study may provide a potential new dual-function stabilizer to be applied to emulsions.

## 2. Results and Discussion

### 2.1. Molecular Simulation and Single-Factor Analysis

The interaction poses and conformational characteristics between VE and β-CD were explored via molecular docking. As shown in Figure 1A, the hydrophobic tail of VE embedded in the wide hydrophobic cavity of β-CD, while, the benzene ring head of VE was exposed outside the cavity. The 20 configurations with the lowest binding energies among all the docking results of β-CD and VE were selected. The lowest binding energy of β-CD and vitamin E was −5.58 kcal/mol, which was the most stable configuration in the docking results of β-CD and VE. The six-membered epoxy ring in the VE molecule was located in the cavity of β-CD, the benzene ring was at the large end of β-CD, and the hydrophobic carbon chain was at the small end of β-CD. The molecular docking results showed that there was no hydrogen bond or hydrophobic interaction between β-CD and VE molecules, but there was a weak van der Waals force. In addition, the lowest binding energy of all docking results for OSA and VE was −2.0 kcal/mol. The molecular docking between OSA and VE suggested that the oxygen atom in the five-membered ring of OSA formed a strong hydrogen bond interaction (1.89 Å) with the hydrogen atom in the hydroxyl group of C-6. A van der Waals force also existed between OSA and VE (Figure 1B). This result was in line with the study of Xi et al. [19]. Therefore, OSA groups were introduced to β-CD to achieve a relatively higher embedding rate of VE. In the single-factor study, the reaction temperature (°C), OCD/VE (w/w) ratio and reaction time (min) were investigated.

The inclusion rate and content of VE increased with the increase of the reaction temperature from 35 to 45 °C, and decreased after the temperature was raised above 45 °C (Figure 2A). As the temperature increased, the solubility of OCD increased. Heating caused the molecules to move faster, which contributed to the formation of the inclusion complex. However, the process of inclusion complex formation was an exothermic process. When the temperature was too high, the VE molecules were able to break loose from the cavity of OCD. The OCD/VE ratio could affect the inclusion rate significantly as well. To study the effect of different OCD/VE ratios on the inclusion rate and content of VE, six OCD/VE ratio points (30:1, 25:1, 20:1, 16:1, 12:1, 10:1) were employed (Figure 2B). It was found that when the OCD/VE ratio decreased from 30: 1 to 12: 1, the content of VE and the inclusion rate increased significantly, while it reduced sharply when the OCD/VE ratio was lower than 12:1. The chances that OCD would come into contact with VE increased with the increase in OCD concentration. Thus, inclusion rate and content of VE increased correspondingly. After the whole system reached a state of dynamic equilibrium, the continuous addition of OCD was not conductive to the inclusion. Reaction time also affected the inclusion rate significantly. Increasing the reaction time beyond 5 h resulted in a decrease in inclusion rate and content of VE. The inclusion complex could be broken up, and VE released free into the system once more, upon stirring for a long time (Figure 2C).

### 2.2. SEM Analysis

The surface morphology of the powders derived from β-CD, OCD, OCD and VE physical mixture, and OCD/VE inclusion complex was assessed by SEM. As shown in Figure 3, β-CD existed in a needle-like crystal, which was rectangular-shaped with tiny fractures appearing at the surface. Meanwhile, OCD was observed as sheet-like aggregate and the surface was seamless. On the other hand, the physical mixture presented a similar morphology to OCD. The OCD/VE inclusion complex appeared in the form of an irregular dense block, with distinct edges and corners in which the original morphology of OCD disappeared and multilayered aggregates of pieces were present. Similar results have been previously reported, namely that the changes in the surface morphology of the crystal provide strong evidence of the formation of the inclusion complex [20,21].

### 2.3. AFM Analysis

AFM scans the surface of samples with a nanoscale probe to produce accurate topographic images. Therefore, AFM is suitable for observing nanometer-scale surface roughness and observing the surface texture of deposited films, especially when the surface feature size is much smaller than 3 μm [22,23]. Both 2D and 3D AFM images of β-CD, OCD, OCD and VE physical mixture, and OCD/VE inclusion complex are shown in Figure 4. After modification with OSA, the particles of β-cyclodextrin were slightly aggregated. Compared with OCD, the surface morphology of the OCD/VE inclusion complex, in which particles were randomly distributed in the field of vision did not change significantly. However, large aggregates appeared in the OCD and VE physical mixture, which showed a square crystal shape. The changes in the surface topography and crystal structure provided strong evidence of the formation of the OCD/VE inclusion complex [24,25]. The average roughness values for β-CD, OCD, OCD and VE physical mixture, and OCD/VE inclusion complex were 25.38, 19.89, 74.03, and 19.48, respectively. The inclusion compound had a lower surface roughness. It was further confirmed that VE was encapsulated in OCD.

### 2.4. FT-IR Analysis

The FT-IR spectra of β-CD, OCD, OCD and VE physical mixture, and OCD/VE inclusion complex are shown in Figure 5. The wavelength from 950 to 1200 cm^−1^ is a fingerprint of carbohydrates, because it recognizes the functional groups of carbohydrates [26]. As shown in the fingerprint region, the discernible absorption peaks at 1155, 1082, and 1032 cm^−1^ were characteristic of the anhydroglucose ring C-O stretching vibration, and the peak at 941 cm^−1^ was attributed to the skeletal mode vibration of the α-(1→4) glycosidic linkage [27]. The extremely broad band at 3415 cm^−1^ was attributed to the stretching vibration of hydroxyl (O-H) groups, and the band at 2924 cm^−1^ was characteristic of the C-H stretching vibration [28]. The absorption peak at 1639 cm^−1^ was the bending vibration peak of water molecules in the β-cyclodextrin cavity [29].

Compared with β-CD, two new absorption peaks at 1722 and 1569 cm^−1^ were observed in OCD, which were attributed to the stretching vibration of the newly formed ester bond and the asymmetric stretching vibration of the carboxyl group, respectively [14]. The FT-IR spectrum of the OCD and VE physical mixture was extremely similar to that of OCD, indicating that the skeleton structure of OCD had not changed in the physical mixture [30]. However, for the OCD/VE inclusion complex, it was clear that the O-H stretching vibration peak of OCD had shifted from 3415 to 3380 cm^−1^, indicating that the conformation of OCD had changed. All the characteristic absorption peaks of VE were obviously weakened, suggesting that VE was already entrapped in the cavity of OCD.

### 2.5. XRD Analysis

XRD is a reliable technique for studying the crystal structure of cyclodextrin and the inclusion complexes of cyclodextrins and guest molecules [31]. The XRD patterns of β-CD, OCD, the physical mixture of OCD and VE, and OCD/VE inclusion complex are shown in Figure 6. Previous studies have shown that cyclodextrin and its inclusion complexes have three crystal structures: cage-type, channel-type, and layer-type [32]. It can be seen from the XRD patterns that β-CD exhibits characteristic diffraction angles (2θ) at 11.2°, 12.8°, 13.4°, and 18.1°, indicating that β-CD exhibits a typical cage-type structure. However, these characteristic diffraction angles (2θ) disappeared in the XRD pattern of OCD, and the characteristic diffraction angles (2θ) of OCD appeared at 10.4°, 12.2°, and 19.3°, indicating that the introduction of the OSA groups changed the crystal structure of β-CD. The physical mixture exhibited characteristic diffraction angles (2θ) similar to that of OCD. In contrast, compared with OCD and the physical mixture, the OCD/VE inclusion complex exhibited a different XRD pattern with characteristic diffraction angles (2θ) at 10.7°, 11.7°, 12.6°, and 17.8°, indicating that the inclusion complex formed a special channel-type structure [33,34].

### 2.6. ^13^C CP/MAS NMR Analysis

The molecular structures of β-CD, OCD, OCD and VE physical mixture, and OCD/VE inclusion complex were further characterized by ^13^C CP/MAS NMR spectra (Figure 7). The peaks of the carbon atoms in the glucopyranose molecule were assigned as follows: C-a (102.49 ppm), C-b, C-c, and C-e (71.37 ppm), C-d (81.20 ppm), and C-f (60.07 ppm) [35]. It is well known that the formation of an inclusion complex can alter the conformation and electromagnetic environment of host and guest molecules, which can be recorded in their ^13^C CP/MAS NMR spectra [36]. As shown in Figure 7, the carbon atoms in the C-d and C-f positions of the β-cyclodextrin molecules showed split signal peaks, which was attributed to the fact that the carbon atoms for all the C-a ~ f positions of the β-cyclodextrin molecules were in a state of mutual resonance, indicating that the β-cyclodextrin molecules were in an asymmetric crystalline state [37]. These resonance signals changed upon the introduction of OSA groups which altered the chemical environment of the carbon atoms of the β-cyclodextrin molecules. In addition, these split signal peaks in the physical mixture disappeared and a new resonance peak appeared at 25.16 ppm, due to the signal peaks at C-6′, C-10′, and C-12′a-CH_3_ positions of the VE molecules [38]. The carbon atoms in the inclusion complex exhibited a chemical environment similar to that of the OCD molecules. Moreover, the signal peaks of the carbon atoms in the physical mixture showed sharp singlets. All of these results indicated that the cyclodextrin molecules exhibited a more symmetrical conformation after the formation of the inclusion complex.

### 2.7. Emulsions Physical Stabilities

The physical stability of the emulsions prepared with β-CD and its derivative were evaluated by particle size distribution (Figure 8A), ζ-potential (Figure 8B), and creaming index (CI, Figure 8C). As shown in Figure 8A, all the particle diameter distributions were measured within 10 μm, which was in the range of distribution previously reported by other studies. However, the droplet size of the β-CD-stabilized emulsion was significantly higher than that of those stabilized by OCD, which was due to the introduction of OSA groups enhancing the surface activity of β-CD. In addition, the smaller droplet size and lower peak width of the β-CD/VE and OCD/VE showed that the introduction of VE was beneficial to the stability of the emulsion. The ζ-potential is an important index used to evaluate the physical stability of a dispersion system. It is an important characteristic parameter that is able to indicate the interaction between charged emulsion droplets. According to the electric double layer theory in an oil-in-water emulsion, the electrophoretic velocity of the dispersed phase was obtained to calculate the ζ-potential. As shown in Figure 8B, the ζ-potential decreased from −37.43 ± 1.86 to −40.67 ± 1.12 mV when β-CD was modified with OSA. This result was due to the carboxyl group on the OS group, which increased the electrostatic repulsive force between emulsion droplet particles. The creaming index (CI) is another method used to evaluate the stability of emulsions. As shown in Figure 8C, all the fresh emulsions were stable; however, they presented different CI values after storage for 7 days. The introduction of the OS group and VE could increase the amount of adsorbed particles on the emulsion droplets and the thicker interfacial film was a physical barrier, effectively preventing the oil droplets from coalescing.

### 2.8. Oxidative Stability in Oil-in-Water Emulsions

All the emulsions were incubated at 50 °C for 0 and 30 days, with the goal of determining differences in lipid oxidation among the emulsions. As shown in Figure 9, all the emulsions had similar TBARS values of about 0.12 mmol/kg oil at beginning (0 day). The TBARS values of all samples increased obviously after storage for 30 days. For the emulsion stabilized by OCD without the antioxidant (VE), the TBARS value was as high as 1.47 ± 0.03 mmol/kg oil. However, the emulsion stabilized by β-CD showed a lower TBARS value (1.38 ± 0.02 mmol/kg oil) when compared with the OCD-stabilized one. This result was due to the smaller oil droplets in the OCD-stabilized emulsion, which were much more susceptible to oxidative deterioration than those in the emulsion with β-CD. The physical addition of VE could restrain the increase of TBARS value, as shown in the physical mixture of OCD and VE-stabilized emulsion (1.09 ± 0.05 mmol/kg oil). Among the four samples, the OCD/VE inclusion complex showed the strongest ability to protect the oil in the emulsion from oxidation. It was indicated that the inclusion complex particles had the ability to protect the oil droplets and reduced the content of lipid oxidation in the emulsion. In general, small oil droplets in emulsions are much more susceptible to oxidative deterioration than bulk oil because of the large contact surface area between oxidizable fatty acids and oxidants [3]. The oil–water interface was the region where the co-oxidant was the most concentrated in the emulsion system. Therefore, lipid oxidation occurred correspondingly at the oil–water interface. In the emulsion stabilized by the OCD/VE inclusion complex, the lipid oxidation product content was the lowest among the four samples. This may have been due to the particles forming an interfacial film on the outer surface of the oil droplets, which prevented contact between the lipid hydroperoxide and the oxygen donor. On the other hand, the natural antioxidant (VE) presented in the clathrate particles released slowly to give longer-term protection for the emulsion.

## 3. Materials and Methods

### 3.1. Materials

β-CD was purchased from Shanghai Yuanye Bio-Technology Co., Ltd. (Shanghai, China) and dried in a vacuum for 24 h. The 2-octen-1-ylsuccinic anhydride (OSA) was purchased from Sigma-Aldrich Chemical Co. (Milwaukee, WI, USA). VE (purity > 97%) was purchased from Aladdin Industrial Inc. (Shanghai, China). Other reagents and chemicals were of analytical reagent grade.

### 3.2. Molecular Docking

The crystal structures of β-CD, OSA (PubChem CID: 5362689), and VE (PubChem CID: 86472) were obtained from the RCSB Protein Data Bank (PDB 200 ID: 3CGT) and PubChem database. Molecular docking (β-CD vs. VE, OSA vs. VE) was performed using AutoDock Tools 4.2 software (Scripps Research Institute, La Jolla, CA, USA). The receptor remained rigid, and the ligand remained flexible. The autogrid box parameter was set to 60 Å × 60 Å × 60 Å, and the grid spacing parameter was 0.375 Å. The calculation was performed using the Lamarckian genetic algorithm (LGA), and other parameters were set to default values [39]. The docking results were analyzed using PyMOL 1.8.x (DeLano Scientific LLC, Palo Alto, CA, USA) and Discovery Studio platform (version 4.5.0, Biovea Inc, Omaha, NE, USA).

### 3.3. Preparation of OCD/VE Inclusion Complex

OCD was prepared under optimized alkaline conditions by esterifying β-CD with OSA, according to our previous method [14]. Briefly, the pH of the β-CD suspension (10%, *w*/*w*) was adjusted to 8.5 with NaOH solution (3%, *w*/*v*), and the temperature was controlled at 50 ± 1 °C. OSA (3%, based on the β-CD dry weight) was added slowly and the reaction was completed when the pH reached a constant value. After the completion of the reaction, the solution was neutralized to pH 6.5 with 3% HCl and freeze-dried. The OCD was obtained by washing the freeze-dried power samples using hexane/isopropanol (3:1 *v*/*v*) five times. OCD (3 g, dry weight) was dissolved in Milli-Q water (10%, *w*/*w*) and stirred for 1 h until the OCD dissolved completely. A certain mass ratio of VE (OCD: VE = 30: 1, 25: 1, 20: 1, 16: 1, 12: 1, 10: 1) was dissolved in isopropyl alcohol and added dropwise to the OCD solution. The inclusion complexes were obtained by stirring the mixture at a certain temperature (35, 40, 45, 50, 55, or 60 °C) for a certain time (2, 3, 4, 5, 6, or 7 h), and were then stored at 4 °C for 12 h under a sealed nitrogen atmosphere. Subsequently, the mixture solution was evaporated to remove the isopropyl alcohol and then freeze-dried. The dried powder samples were washed with isopropyl alcohol to remove the residual VE. The absorbance of the supernatant was measured with a UV-vis spectrophotometer (TU 1810 SPC) at 292 nm. The VE content was calculated using a standard curve. The inclusion rates (X, %) of VE were calculated using the following equation:(1)X=(B−C×N×V)/B×100%
where B (mg) is the total mass of added VE, C (mg/mL) is the VE concentration of the washing liquid (after being diluted) determined from the VE standard curve, N is the dilution ratio of the washing liquid, and V (mL) is the total volume of the washing liquid.

### 3.4. Scanning Electron Microscopy (SEM)

The morphology of β-CD, OCD, OCD/VE inclusion complex, and the physical mixture of OCD and VE were assessed using scanning electron microscopy (Quanta 250, FEI Co., Hillsboro, OR, USA). Prior to examination, samples were prepared by mounting about 0.5 mg of powder onto a 5 × 5 mm silicon wafer affixed via graphite tape to an aluminum stub. The powder was then sputter-coated for 40 s at a beam current of 38–42 mA with a 200 layer of gold/palladium alloy. An accelerating potential of 50.0 kV and an Everhart-Thornley detector (ETD) detector was used.

### 3.5. Atomic Force Microscopy (AFM)

Ten microliters of solution of β-CD, OCD, OCD/VE inclusion complex, or the physical mixture of OCD and VE (10 μg/mL) was deposited on the surface of a cut mica plate, dried in a desiccator, and captured in tap mode. The AFM image of the sample was captured using an atomic force microscope (EKYS-121, Veeco, Santa Barbara, CA, USA). The resonance frequency was set at 300 kHz and the force constant was 40 N/m.

### 3.6. Fourier-Transform Infrared Spectroscopy (FT-IR)

The dried powders were blended with potassium bromide, pressed into tablets, and examined using a Fourier-transform infrared spectrometer (FTIR) (TENSOR27, Bruker, Germany) equipped with a deuterated triglycine sulfate detector (DTGS) in the range of 4000~500 cm^−1^ at a resolution of 4 cm^−1^ [40].

### 3.7. X-Ray Diffraction (XRD)

X-ray diffraction spectra were recorded on a D8 advance powder XRD diffractometer (Bruker, Germany) with a copper target X-ray tube set at 40 kV. The measurement angle 2θ range was 5° to 50°, the scan step length was 0.04 degrees, and the scan speed was a 38.4 s/step. The powder samples were placed in a rectangular aluminum cell and measured at ambient temperature.

### 3.8. ^13^C CP/MAS NMR

Lyophilized powder samples were packed into 4 mm zirconia rotors and then measured directly using a ^13^ C CP/MAS NMR spectrometer (Avance 400 MHz, Bruker, Germany). The field strength was 9.40 T, the speed was 15 kHz, the pulse width was 90°, the cross-polarization time was 4 μs, the contact time was 2 ms, and the sampling time was 34 ms. The sampling interval was 2 s, the number of scans was 1024, and the spectral width was 300 ppm [10].

### 3.9. Emulsion Preparation

β-CD, OCD, OCD/VE inclusion complex, and β-CD/VE inclusion complex (1.5 g, dry weight) were suspended in Milli-Q water (88.5 g) with stirring at room temperature for 24 h to obtain a solution. The solutions (90 g) were then mixed with camellia oil to achieve a final mass of 100 g. The fine emulsion was pre-homogenized using an Ultra-Turrax device (T18 basic, IKA, Staufen, Germany) at 24,000 rpm for 3 min, and then homogenized through an ultra-high-pressure homogenizer (Nano DeBEE, BEE International Inc., South Easton, MA, USA) operated at 30 MPa. After homogenization, sodium azide (0.01%, *w*/*w*) was added to the emulsions to inhibit the growth of microorganisms.

### 3.10. ζ-Potential Measurements

The **ζ**-potential was measured via a phase analysis light-scattering technique using a Malvern Zetasizer (Nano ZS90, Malvern Instruments, Worcestershire, UK). Emulsion samples were diluted 1000-fold with Milli-Q water prior to measurement to avoid any multiple scattering effect. The diluted emulsions were mixed thoroughly and then injected into the sample cell, in which the temperature was maintained at 25 °C. The ζ-potential of each sample was calculated from the average of triplicate measurements on the diluted emulsion.

### 3.11. Particle Size Distribution Measurements

According to the method reported by Li, Fang, Al-Assaf, Phillips, and Jiang [41], a Malvern Mastersizer 2000 (Zetasizer Nano-ZS, Malvern Instruments, Worcestershire, UK) was used to determinate the particle size distribution of the emulsion. Briefly, a few drops of the emulsion samples were dispensed in the water in the measuring cell of the instrument with stirring at 2000 rpm until the obscuration degree was approximately 15%, and the refractive indices of camellia oil and water were set to 1.47 and 1.33, respectively.

### 3.12. Creaming Stability Measurement

A 15 mL of the freshly prepared emulsion was injected into the vial and stored at room temperature for 1 week to observe the stratification of the emulsion. The creaming index (CI, %) was calculated as follows:CI = (Height of the serum layer/Total height of the emulsion) × 100%(2)

### 3.13. Emulsion Oxidative Stability

The degree of lipid oxidation was assessed via 2-thiobarbituric acid reactive substances (TBARSs). The emulsions were placed in tightly sealed screw-cap test tubes and incubated at 50 °C in the dark for up to 30 days to allow lipid autoxidation. TBARSs were tested according to the method reported by Mei, Decker, and McClements [7] and were calculated from the standard curve prepared using 1,1,3,3-tetraethoxypropane. Briefly, a TBA solution was prepared by mixing 15 g of trichloroacetic acid, 0.375 g of TBA, 1.76 mL of HCl (12 N), and 82.9 mL of H_2_O. TBA solution (100 μL) was mixed with 3 mL of 2% butylated hydroxytoluene in ethanol, and 2 mL of this solution was mixed with 0.3 mL of the emulsion and 0.7 mL of H_2_O. The mixture was then heated in a boiling water bath for 15 min, cooled to room temperature using tap water, and centrifuged at 2000 g for 15 min. The absorbance was measured at 532 nm.

### 3.14. Statistical Analysis

All tests were performed in triplicate. Analysis of variance was performed, and results were evaluated using the Tukey–Kramer multiple comparison test (*p* < 0.05) using the SPSS 17.0 statistical software (SPSS Inc., Chicago, IL, USA). Origin (Origin Lab Co., Pro.8.0, Northampton, MA, USA) software was used for data processing and to create charts.

## 4. Conclusions

In this study, β-CD was modified with OSA and the esterification product (OCD) was used to prepare an OCD/VE inclusion complex. In addition, the structure of the OCD/VE inclusion complex was characterized using FT-IR, XRD, ^13^C CP/MAS NMR, AFM/SEM, and molecular docking. The results of the FT-IR analysis suggested that VE was entrapped into the cavity of OCD. Compared with OCD and the physical mixture, the OCD/VE inclusion complex exhibited a special channel-type XRD pattern with characteristic diffraction angles (2θ) at 10.7°, 11.7°, 12.6°, and 17.8°. The physical stability of the emulsions stabilized with OCD and its physical mixture and inclusion complex were evaluated by particle size distribution, ζ-potential, and CI. Due to the introduction of OSA groups, the droplet size of the β-CD-stabilized emulsion was significantly higher than that of those stabilized by OCD. In addition, the smaller droplet size and lower peak width of the OCD/VE showed that the introduction of VE was also beneficial to the stability of the emulsion. Among all the samples, the OCD/VE inclusion complex showed the strongest ability to protect the oil droplets and reduced the content of lipid oxidation in the emulsion.

## Figures and Tables

**Figure 1 molecules-25-00654-f001:**
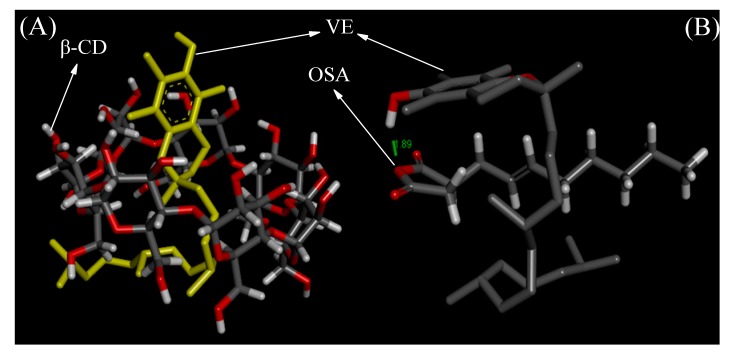
Molecular docking of β-cyclodextrin (β-CD) vs. vitamin E (VE) (**A**) and octenyl succinic anhydride (OSA) vs. VE (**B**).

**Figure 2 molecules-25-00654-f002:**
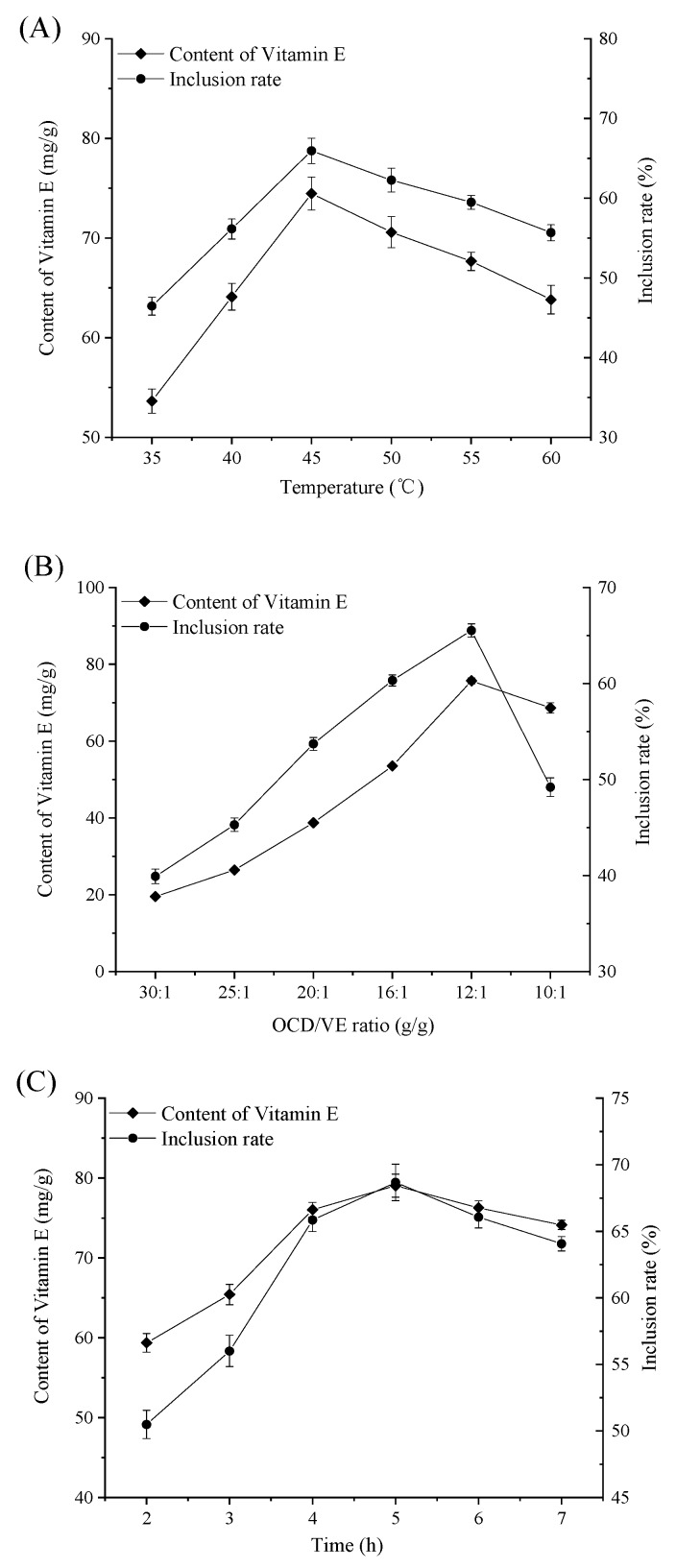
Effects of reaction temperature (**A**), octenyl succinic-β-cyclodextrin (OCD)/VE ratio (**B**), and reaction time (**C**) on the inclusion rate and content of VE.

**Figure 3 molecules-25-00654-f003:**
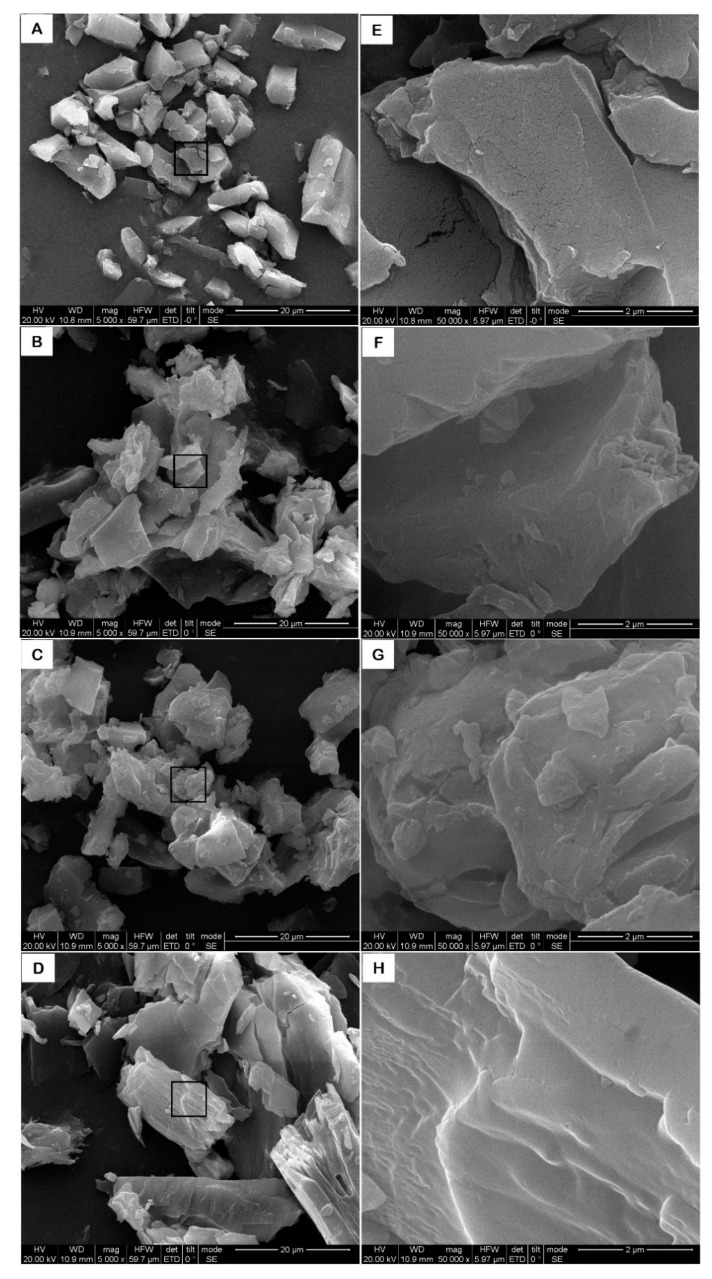
Scanning electron microscopy (SEM) images of β-CD (**A**,**E**), OCD (**B**,**F**), OCD and VE physical mixture (**C**,**G**), and OCD/VE inclusion complex (**D**,**H**). The magnification in panels (**A**–**D**) is 5000× and in panels (**E**–**H)** is 50,000×.

**Figure 4 molecules-25-00654-f004:**
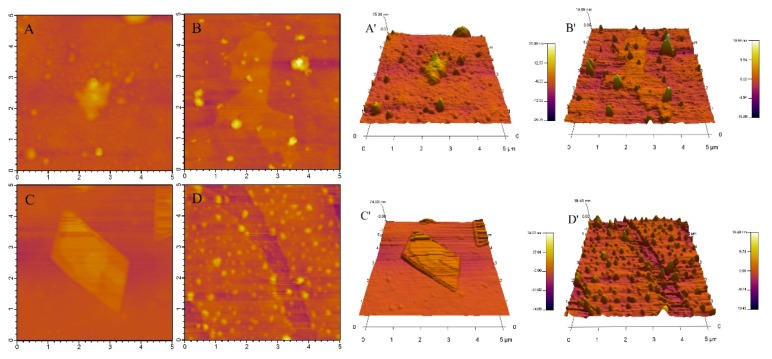
2D and 3D atomic force microscopy (AFM) images (5 μm × 5 μm) of β-CD (**A**,**A’**), OCD (**B**,**B’**), OCD/VE inclusion complex (**C**,**C’**), and OCD and vitamin E physical mixture (**D**,**D’**).

**Figure 5 molecules-25-00654-f005:**
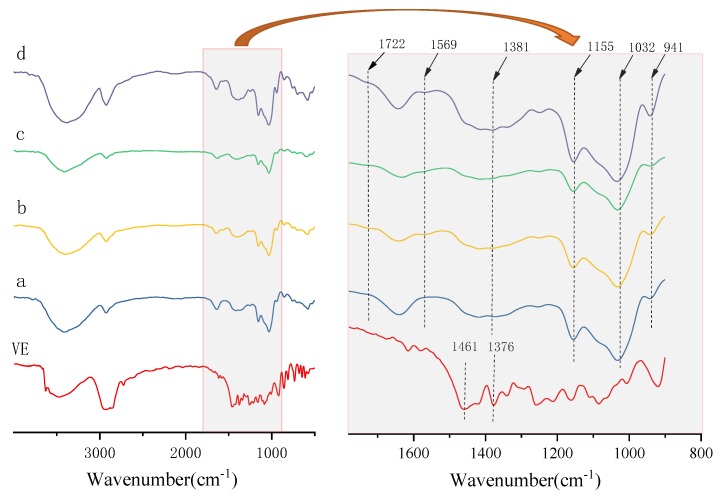
Fourier-transform infrared spectroscopy (FT-IR) spectra of β-CD (**a**), OCD (**b**), OCD and VE physical mixture (**c**), and OCD/VE inclusion complex (**d**).

**Figure 6 molecules-25-00654-f006:**
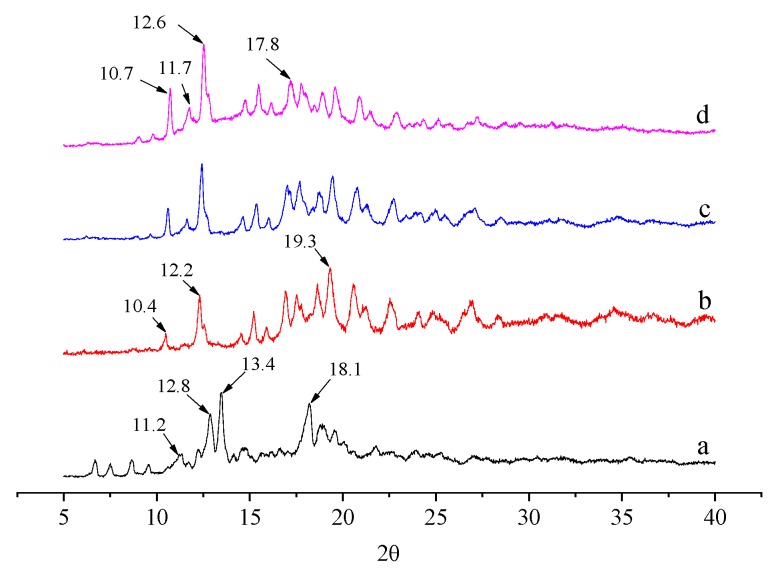
X-ray diffraction (XRD) patterns of β-CD (**a**), OCD (**b**), OCD and VE physical mixture (**c**), and OCD/VE inclusion complex (**d**).

**Figure 7 molecules-25-00654-f007:**
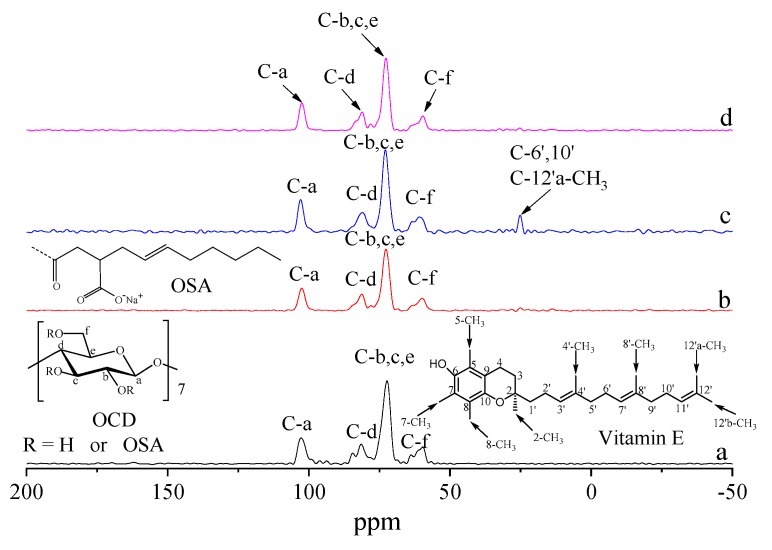
^13^C CP/MAS NMR spectra of β-CD (**a**), OCD (**b**), OCD and VE physical mixture (**c**), and OCD/VE inclusion complex (**d**).

**Figure 8 molecules-25-00654-f008:**
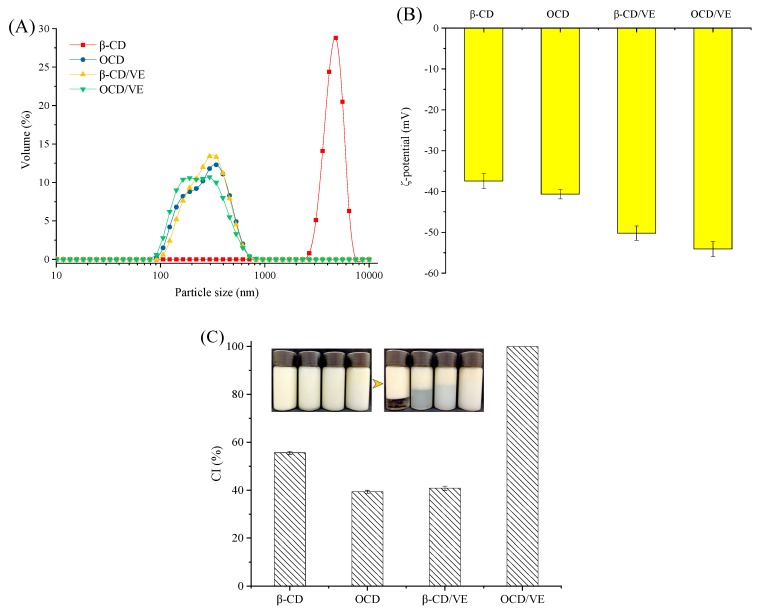
Particle size distribution (**A**), ζ-potential (**B**), and creaming index (**C**) of the emulsions stabilized by β-CD, OCD, β-CD/VE inclusion complex, and OCD/VE inclusion complex. Phase separation profiles of emulsions (**C**) after 0 days (left) and 7 days (right) of storage at 25 °C.

**Figure 9 molecules-25-00654-f009:**
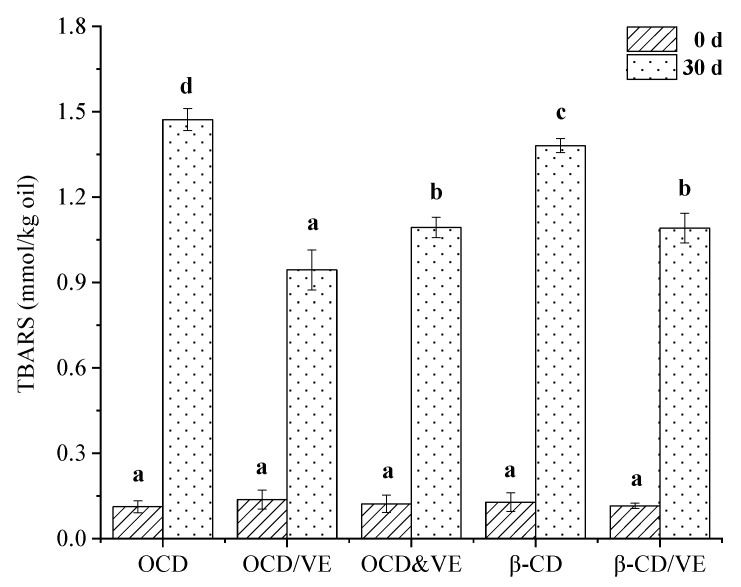
TBARS values of the emulsions stabilized by OCD, physical mixture of OCD and VE (OCD&VE), inclusion complexes of OCD/VE, β-CD, and β-CD/VE at 50 °C. The different letters (**a**–**d**) above each bar indicate significant differences at *p* < 0.05.

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
