# Peer review of "Preparation and Characterization of Octenyl Succinate β-Cyclodextrin and Vitamin E Inclusion Complex and Its Application in Emulsion"

_molecules, 2020, doi:10.3390/molecules25030654_

Round 1
Reviewer 1 Report
The manuscript “Preparation and Characterization of Octenyl 2 Succinate β-Cyclodextrin and VE Inclusion Complex and Its Application in Emulsion” by Dongmei Ke et al. characterizes the physicochemical properties of the Octenyl 2 Succinate β-Cyclodextrin (OCD) and Vitamin E (VE) Inclusion Complex. This OCD/VE complex was prepared to increase the interaction of the antioxidant VE with b-cyclodextrin (b-CD) to protect oil emulsions against oxidation. The structure of the OCD/VE inclusion complex was characterized by AFM/SEM, FT-IR, XRD and 13C CP/MAS NMR and it was compare with the structure of b-CD, OCD and OCD and VE mixture showing that VE was efficiently entrapped in the OCD cavity. Moreover, the physical stability of camellia oil emulsions with OCD/VE showed that the OCD/VE complex was beneficial to the stability of emulsion. The oxidative stability of the OCD/VE emulsion was also shown to be higher than the stability of b-CD, OCD or OCD and VE mixture. Thus, this manuscript describes an exhaustive characterization of OCD/VE inclusion complex and emulsion and it is suitable for publication in molecules.
However there some issues that should be solved before publication:
-In general, the authors should improve the clarity, language and style of the manuscript.
-Title, line 3. VE should be unabbreviated (Vitamin E).
Please use VE (and not V E) in the entire manuscript.
-Abstract, There is some confusion between β-cyclodextrin and octenyl succinic-β-cyclodextrin (OCD). Could be better something like: line 19 “β-CD was modified using OSA to produce octenyl succinic-β-cyclodextrin (OCD)” and then line 20. “inclusion complexes were prepared using β-CD and OCD with VE”. Same in line 24 “VE had been embedded into the cavity of OCD”.
-Autodock. Please provide the binding energies calculated by autodock. Please discuss the different docking models obtained (e.g. different orientations) and a model of the OCD/VE complex.
-line 102. “the OCD/VE ratio increased from 30:1 to 12:1”. From 30:1 to 12:1 there is a decrease. In any case “the VE/OCD ration increased from 1:30 to 1:12”. Please clarify.
-Figure 3. Red labels in figure 3 are difficult to read. Which is the magnification in panels A,B,C and D and in panels E,F,G,H?
-Figures 5,6 and 7. Please show with arrowheads or symbols the peaks you highlight in the manuscript.
-Figure 8. Please write in the figure legend the different experiments shown. Particle size distribution (A), ζ-potential (B) and creaming index (C).
-Figure 9. Please write in the figure legend the meaning of a, b and c
Author Response
The manuscript “Preparation and Characterization of Octenyl Succinate β-Cyclodextrin and VE Inclusion Complex and Its Application in Emulsion” by Dongmei Ke et al. characterizes the physicochemical properties of the otenyl sccinate β-cyclodextrin (OCD) and vitamin E (VE) inclusion complex. This OCD/VE complex was prepared to increase the interaction of the antioxidant VE with β-cyclodextrin (β-CD) to protect oil emulsions against oxidation. The structure of the OCD/VE inclusion complex was characterized by AFM/SEM, FT-IR, XRD and 13C CP/MAS NMR and it was compared with the structure of β-CD, OCD and OCD and VE mixture showing that VE was efficiently entrapped in the OCD cavity. Moreover, the physical stability of camellia oil emulsions with OCD/VE showed that the OCD/VE complex was beneficial to the stability of emulsion. The oxidative stability of the OCD/VE emulsion was also shown to be higher than the stability of β-CD, OCD or OCD and VE mixture. Thus, this manuscript describes an exhaustive characterization of OCD/VE inclusion complex and emulsion and it is suitable for publication in molecules.
However there some issues that should be solved before publication:
-In general, the authors should improve the clarity, language and style of the manuscript.
-Title, line 3. VE should be unabbreviated (Vitamin E).
Please use VE (and not V E) in the entire manuscript.
Response: It has been revised according to your suggestion. (line 27, 32)
-Abstract, there is some confusion between β-cyclodextrin and octenyl succinic-β-cyclodextrin (OCD). Could be better something like: line 19 “β-CD was modified using OSA to produce octenyl succinic-β-cyclodextrin (OCD)” and then line 20. “inclusion complexes were prepared using β-CD and OCD with VE”. Same in line 24 “VE had been embedded into the cavity of OCD”.
Response: We all think well of your suggestion and the abstract has been revised accordingly. (line 19-23) -Autodock.
Please provide the binding energies calculated by autodock.
Please discuss the different docking models obtained (e.g. different orientations) and a model of the OCD/VE complex.
Response: We selected the 20 configurations with the lowest binding energy among all the docking results of β-CD and vitamin E.
The lowest binding energy of β-CD and vitamin E is -5.58 kcal/mol, which is the most stable configuration in the docking results of β-CD and vitamin E. The six-membered epoxy ring in the vitamin E molecule is located in the cavity of β-CD, the benzene ring is at the large end of β-CD, and the hydrophobic carbon chain is at the small end of β-CD.
The lowest binding energy of all docking results of OSA and vitamin E is -2.0 kcal/mol. The hydrogen atom on the hydroxyl of vitamin E molecule forms a hydrogen bond with the oxygen atom in the five-membered ring of OSA molecule. OSA modified β-CD (OCD), introduces OSA group at 6-OH, that is, introduces hydrophilic carboxyl group and hydrophobic carbon chain to improve the surface activity of β-CD and increase the β-CD emulsifying ability.
In addition, since OSA molecules can form hydrogen bonds and van der Waals force with vitamin E molecules, which has a positive effect on β-CD loaded vitamin E. (line 93-104) -line 102. “the OCD/VE ratio increased from 30:1 to 12:1”. From 30:1 to 12:1 there is a decrease. In any case “the VE/OCD ration increased from 1:30 to 1:12”. Please clarify.
Response: The whole manuscript has been checked carefully and the errors have been corrected. (line 118-120) -Figure 3. Red labels in figure 3 are difficult to read. Which is the magnification in panels A, B, C and D and in panels E, F, G, H?
Response: According to your suggestion, red labels in figure 3 have been replaced with white ones and the font size was also enlarged.
In addition, the magnification in panels of A, B, C and D and in panels of E, F, G, H has been added. (line 555-556 and Fig. 3)
-Figures 5, 6 and 7. Please show with arrowheads or symbols the peaks you highlight in the manuscript.
Response: It has been revised according to your suggestion. (Figures 5, 6 and 7)
-Figure 8. Please write in the figure legend the different experiments shown. Particle size distribution (A), ζ-potential (B) and creaming index (C).
Response: It has been revised according to your suggestion. (line 573-576) -Figure 9.
Please write in the figure legend the meaning of a, b and c.
Response: The meaning of a, b and c has been added in the figure legend. (line 579-581)
Reviewer 2 Report
In this paper, the authors described the complexation of vitamin E and beta-cyclodextrin octenyl succinate . The inclusion complex was fully characterized using different techniques and computer modeling was used to establish the host-guest interaction. Finally, they prepared oil/water emulsion and assayed the influence of the inclusion complex on sizes, zeta potential and physical stability of the emulsion.
The work is largely treated and design was appropriate, however, some minor revision must be performed before to accept the paper for publication.
1) the authors should write in the title and Keywords the complete name of Vitamin E and not the acronym;
2)line 79 and 74: the reference n. 3 is incorrect. Please change it
3) the label in figure 1 must be correct. In yellow is showed VE and not OSA
4)SEM analysis: In my opininion SEM analysis is not suitable to evidence the presence of an inclusion complex. The variation in morfology of VE/OCD could be due to the freeze-drying process. All compared samples should be subject to freeze-dried before the analysis.
5) The label of figure 5B or the text must be corrected (1569 cm-1 or 1572cm-1?)
6) The caption of figure 8 should be rewritten more clearly. Furthermore, the time in which was performed the second observation (pictures in the inset of figure 8C) should be reported.
7) Line 227: Please correct o with 0.
8)Figure 9 must be implemented reporting the results for beta-Cyd and Beta-cyd/VE inclusion complex.
9) lines 263-277: the entire paragraph should be rewritten more clearly.
The synthesis of OCD should be reported briefly. The solubility of beta cyclodextrin octenyl succinate should be reported.
The ratio OCD:VE is a molar ratio? Please clarify.
In my opinion, after the evaporation of isopropyl alcohol the authors obtained the complex in solution and a suspension of not included VE. In fact, the authors must wash the freeze-dried powder with the isopropyl alcohol to eliminate the excess of VE. Why the authors do not filter the aqueous solution before the freeze-drying?
10) line 308: what is the volume of water?
11)line 308,309: the autors obtained a dispersion or a solution? Please clarify.
Author Response
List of changes (molecules-705502)
Title: Preparation and Characterization of Octenyl Succinate β-Cyclodextrin and Vitamin E Inclusion Complex and Its Application in Emulsion
We have studied reviewer’s comments carefully and have tried our best to revise our manuscript according to the comments. The detail changes according to the reviewer are list as following:
In this paper, the authors described the complexation of vitamin E and beta-cyclodextrin octenyl succinate. The inclusion complex was fully characterized using different techniques and computer modeling was used to establish the host-guest interaction. Finally, they prepared oil/water emulsion and assayed the influence of the inclusion complex on sizes, zeta potential and physical stability of the emulsion.
The work is largely treated and design was appropriate, however, some minor revision must be performed before to accept the paper for publication.
1) The authors should write in the title and Keywords the complete name of Vitamin E and not the acronym.
Response: It has been revised according to your suggestion. (line 2, 32)
2) line 79 and 74: the reference n. 3 is incorrect. Please change it
Response: The whole references in the manuscript have been checked carefully and revised according to your suggestion. (line 76, 81)
3) the label in figure 1 must be correct. In yellow is showed VE and not OSA
Response: It has been revised according to your suggestion. (Fig. 1)
4) SEM analysis: In my opinion, SEM analysis is not suitable to evidence the presence of an inclusion complex. The variation in morfology of VE/OCD could be due to the freeze-drying process. All compared samples should be subject to freeze-dried before the analysis.
Response: We all think well of your suggestion and the inaccurate expression has been deleted.
5) The label of figure 5B or the text must be corrected (1569 cm-1 or 1572cm-1?)
Response: 1569 cm-1 is correct and it has been revised according to the figure 5. (line 168)
6) The caption of figure 8 should be rewritten more clearly. Furthermore, the time in which was performed the second observation (pictures in the inset of figure 8C) should be reported.
Response: The caption of figure 8 has been rewritten. In addition, the second observation time (7 days) has been added. (line 233-234, 363-367, 573-576 and Fig. 8)
7) Line 227: Please correct o with 0.
Response: It has been revised according to your suggestion. (line 239)
8)Figure 9 must be implemented reporting the results for beta-Cyd and Beta-cyd/VE inclusion complex.
Response: The results for β-CD and β-CD/VE inclusion complex have been added in Figure 9. The discussion was also added in the manuscript. (line 244-248, Fig. 9)
9) lines 263-277: the entire paragraph should be rewritten more clearly.
The synthesis of OCD should be reported briefly. The solubility of beta cyclodextrin octenyl succinate should be reported.
The ratio OCD:VE is a molar ratio? Please clarify.
In my opinion, after the evaporation of isopropyl alcohol the authors obtained the complex in solution and a suspension of not included VE. In fact, the authors must wash the freeze-dried powder with the isopropyl alcohol to eliminate the excess of VE. Why the authors do not filter the aqueous solution before the freeze-drying?
Response: This paragraph has been rewritten and the synthesis of OCD has been added. A certain mass ratio of VE (OCD: VE = 30: 1, 25: 1, 20: 1, 16: 1, 12: 1, 10: 1) was dissolved in isopropyl alcohol and added dropwise to the OCD solution. The inclusion complexes were obtained by stirring the mixture at a certain temperature (35, 40, 45, 50, 55 and 60℃) for a certain time (2, 3 ,4, 5, 6, 7 h), and then stored at 4°C for 12 h under the sealed and nitrogen atmosphere. Subsequently, the mixture solution was evaporated to remove the isopropyl alcohol and then freeze-dried. The dried power samples were washed with isopropyl alcohol to remove the residual VE. (line 284-301)
10) line 308: what is the volume of water?
Response: The volume of water has been added according to your suggestion. (line 340)
11) line 308,309: the authors obtained a dispersion or a solution? Please clarify.
Response: β-CD, OCD, OCD/VE inclusion complex, and β-CD/VE inclusion complex (1.5 g, dry weight) were suspended in Milli-Q water (88.5 g) with stirring at room temperature for 24 h to obtain a solution. (line 341)